# Mild Heat Treatment and Biopreservatives for Artisanal Raw Milk Cheeses: Reducing Microbial Spoilage and Extending Shelf-Life through Thermisation, Plant Extracts and Lactic Acid Bacteria

**DOI:** 10.3390/foods12173206

**Published:** 2023-08-25

**Authors:** Beatriz Nunes Silva, José António Teixeira, Vasco Cadavez, Ursula Gonzales-Barron

**Affiliations:** 1Centro de Investigação de Montanha (CIMO), Instituto Politécnico de Bragança, Campus de Santa Apolónia, 5300-253 Bragança, Portugal; vcadavez@ipb.pt (V.C.); ubarron@ipb.pt (U.G.-B.); 2Laboratório Associado para a Sustentabilidade e Tecnologia em Regiões de Montanha (SusTEC), Instituto Politécnico de Bragança, Campus de Santa Apolónia, 5300-253 Bragança, Portugal; 3CEB—Centre of Biological Engineering, University of Minho, 4710-057 Braga, Portugal; jateixeira@deb.uminho.pt; 4LABBELS—Associate Laboratory, 4710-057 Braga, Portugal

**Keywords:** natural preservatives, inactivation, antimicrobial, dairy, food safety, pathogens

## Abstract

The microbial quality of raw milk artisanal cheeses is not always guaranteed due to the possible presence of pathogens in raw milk that can survive during manufacture and maturation. In this work, an overview of the existing information concerning lactic acid bacteria and plant extracts as antimicrobial agents is provided, as well as thermisation as a strategy to avoid pasteurisation and its negative impact on the sensory characteristics of artisanal cheeses. The mechanisms of antimicrobial action, advantages, limitations and, when applicable, relevant commercial applications are discussed. Plant extracts and lactic acid bacteria appear to be effective approaches to reduce microbial contamination in artisanal raw milk cheeses as a result of their constituents (for example, phenolic compounds in plant extracts), production of antimicrobial substances (such as organic acids and bacteriocins, in the case of lactic acid bacteria), or other mechanisms and their combinations. Thermisation was also confirmed as an effective heat inactivation strategy, causing the impairment of cellular structures and functions. This review also provides insight into the potential constraints of each of the approaches, hence pointing towards the direction of future research.

## 1. Introduction

Cheese is a highly nutritious food, with hundreds of varieties that have different colours, odours, flavours and textures, depending on the type of milk used, production and maturation processes, and age, for example [1].

Artisanal raw milk cheeses are particularly appreciated for their unique sensorial characteristics (namely, texture, aroma and flavour) when compared to other types of cheeses, and there has been a growing demand for specialty and artisanal cheeses due to the number of consumers who currently prefer minimally processed foods that provide a feeling of authenticity [2].

The unique sensorial characteristics of artisanal raw milk cheeses result, among other factors, from the use of unpasteurised milk. In fact, despite having numerous advantages, such as reducing the bacterial load and extending the shelf-life of milk, pasteurisation causes, among other heat-induced changes, denaturation of whey proteins and complex interactions among denatured whey proteins, casein micelles, minerals and fat globules [3]. These modify the biochemistry and microbiology of milk acidification and cheese ripening, and, consequently, the characteristic flavour, aroma and texture of raw milk cheeses cannot be achieved using pasteurised milk [4].

Nonetheless, consumption of raw milk cheeses may pose health safety issues due to the possible presence of pathogenic bacteria in raw milk that can remain viable during manufacture and through ripening [5,6,7]. The consumption of this type of dairy product has caused a few outbreaks [8,9,10,11,12], thus highlighting the need for preservation strategies to improve the microbial safety of raw milk cheeses.

Chemical preservatives would not be suitable for artisanal cheeses, as they would disregard the appeal of a traditional product derived from cultural heritage and produced using only natural, healthy ingredients. Furthermore, they would be an outdated preservation strategy, as the mishandling and extensive consumption of some chemical additives have been shown to induce gut microbiota dysbiosis, which is a contributing factor to various diseases, including neurodegenerative ones [13,14,15]. Finally, current consumer expectations are increasingly towards “clean-label”, chemical preservative-free food products, and consequently, the food industry and scientific community are compelled to investigate novel food preservation methods [16].

Between other techniques, advanced non-thermal technologies (high pressure, cold plasma, pulsed light, and ultrasound) and packaging systems (bioactive films, coating, and modified atmospheric packaging) are among the innovative cheese preservation approaches developed to inactivate microorganisms in milk and extend the shelf life of raw milk cheeses [17]. However, these are not easily implementable for artisanal producers, mainly because of the need for specific and costly equipment, as well as the need for training to operate such technologies.

On the other hand, the incorporation of natural antimicrobial agents in artisanal cheese production is more feasible since starter cultures (lactic acid bacteria, LAB), plant extracts, essential oils and propolis [17,18] can be easily purchased and added directly to the milk, cheese curd, or final product. Another alternative would be to implement a mild thermal process such as thermisation, which uses sub-pasteurisation temperatures to reduce bacterial load while avoiding large heat-induced changes in milk that would affect the final typical organoleptic characteristics of raw milk cheeses [19,20]. This technology would also be easy for artisanal producers to implement since it does not require specialised equipment. Figure 1 displays a schematic diagram of the general cheesemaking process, and the steps at which thermisation, addition of plant extracts, and addition of a starter culture may be implemented are highlighted.

Considering the above-mentioned possibilities, this review presents an overview of the existing information on LAB and plant extracts as biopreservative strategies, as well as thermisation as a mild heat treatment, to be used in raw milk cheeses.

The main microorganisms involved in cheese spoilage are described, and for each biopreservation strategy, the various targets, mechanisms of antimicrobial action, limitations and, when applicable, relevant commercial applications are discussed.

## 2. Spoilage Microorganisms in Raw Milk and Raw Milk Cheeses

The most prevalent spoilage fungi genera identified in raw milk and cheeses are *Candida*, *Cryptococcus*, *Debaryomyces*, *Geotrichum*, *Kluyveromyces*, *Trichosporon*, *Pichia*, and *Rhodotorula* spp. (yeasts), and *Penicillium*, *Aspergillus*, *Cladosporium*, *Mucor*, *Fusarium* and *Alternaria* spp. (moulds) [21,22,23,24]. *Candida rugosa*, *Geotrichum candidum*, *Torulaspora delbrueckii*, *Kluyveromyces marxianus* and *Yarrowia lipolytica* are among the common yeast species found in raw milk, while *Penicillium commune* is one of the most frequently occurring mould species [22,24,25].

In the case of excessive yeast growth, cheese softening and unpleasant flavours and odours may occur, as well as the formation of brown spots on the cheese surface (caused by *Y. lipolytica*, for example) and cheese blowing, caused by early gas formation (as a result of high counts of *T. delbrueckii*, for instance) [22]. Yeasts are also able to promote an increase in the pH of the cheese surface, enabling the growth of pathogenic bacteria [22]. Moulds can also produce off-flavours and undesired pigments, as well as synthesise toxic metabolites, such as mycotoxins [23]. 

Psychrotrophic bacteria dominate the microflora of raw milk, particularly species of the genera *Pseudomonas*, *Acinetobacter*, *Aeromonas*, *Serratia*, *Bacillus*, *Lactococcus*, *Microbacterium*, and *Staphylococcus* [21,26]. Other bacteria associated with cheese spoilage are *Enterobacteriaceae* and clostridial species (*E. cloacae*, *E. agglomerans*, *E. zakazakii*, C. *tyrobutyricum*, *C. butyricum*, *C. sporogenes*, and *C. beijerinckii* have been isolated from milk) [27,28]. Clostridial species may produce excessive amounts of gas and butyric acid during growth, resulting in cheese blowing [27]. Likewise, some *Enterobacteriaceae* species can also cause early blowing in cheese, as well as negatively affect the organoleptic features of cheeses [29]. Moreover, the *Enterobacteriaceae* family includes a wide range of pathogenic bacteria.

The main pathogens of concern that have been detected in raw milk cheeses include enterotoxin-producing *Staphylococcus aureus*, Shiga toxin-producing *Escherichia coli* (STEC; *E. coli* O157:H7, for example), *L. monocytogenes*, *Salmonella* spp., *Brucella* spp. and *Campylobacter* spp. [2,6,10,21,30]. These pathogens may be shed directly into milk via the udder by a diseased or infected animal or may enter milk from the external surfaces of animals, the environment, the milking environment, equipment or from personnel (operators’ hands, for example) [6,30]. *L. monocytogenes* and STEC have been identified as especially high-risk pathogens owing to the severity of illness and potential lethality associated with each [19].

To reduce spoilage in dairy products, adequate cleaning and sanitation of the processing environment is imperative, but on-farm interventions (to reduce the concentration of spores and pathogens in bulk tank raw milk) and processing technologies (such as bactofugation and microfiltration) may also be used [31]. However, in the context of artisanal cheese production, the referred processing technologies are generally not used.

## 3. Biopreservation Strategies

### 3.1. Plant Extracts

The use of plants and herbs as colouring and flavouring agents in cheese manufacture is not new, with some traditional herb-flavoured cheeses having centuries of history [32]. However, plants may be used for more than their organoleptic and decorative properties, owing to their phytochemical constituents that have been shown to have antimicrobial activity [18,33]. The addition of plants and herbs to cheese can be carried out by incorporating them into milk (before cheese making), into cheese curd, or by rolling the cheese into crushed herbs, for example [18].

Plant extracts can be obtained from a multitude of plants using various solvents and extraction methodologies. However, if intended for human consumption, they must be obtained using non-toxic solvents authorised for the industrial production of foodstuffs and food ingredients [34], such as water, ethanol, or their combination.

Conventional extraction procedures include maceration, percolation, infusion, decoction, reflux extraction, Soxhlet extraction and hydro-distillation (which can be subcategorised into steam-, water-distillation, or a combination of both) [35,36,37]. While these may still be widely used, nowadays, it is crucial to consider the ecological impact of extraction methods and those that are more sustainable and “green”, reducing the amount of solvents used and waste generated, and optimising the recovery of bioactive compounds with high added value, should be preferred [38]. To this, techniques such as subcritical water extraction, supercritical fluid extraction, enzyme-, microwave- and ultrasound-assisted extractions, pulsed electric field extraction and accelerated solvent extraction can be used, among many other modern procedures [35,36,37,39]. Moreover, as the extraction method, temperature, solvent and pressure, for example, influence the chemical profile of the extracts produced, the most appropriate extraction parameters should be selected, considering the desired compounds and bioactivity [38,40]. In addition, the plant genotype, geographical location, and environmental and agronomic conditions, among other factors, also contribute to variations in the chemical composition of plant extracts [41].

Based on their structure, plant-derived chemicals may be classified as alkaloids, organosulfur compounds, phenolic compounds, coumarins and terpenes [42]. Generally, phenolic compounds are found in higher concentrations in plants [43] and are assumed to be the main antimicrobial agents [43,44,45], although the remaining compounds have also shown this capacity [42]. With respect to the chemical structure of the bioactive compounds, it has been demonstrated that functional groups such as hydroxyl groups and the number of double bonds can influence antimicrobial strength [37].

The exact targets of plant antimicrobials are often difficult to define, considering the many interacting reactions taking place simultaneously [33] and the various compounds found in plant extracts, each exerting its own effect [42]. Nonetheless, several mechanisms have been suggested to explain the antimicrobial mode of action of plant extracts. These include inhibition of efflux pumps (implicated in the export of harmful substances from within the cell into the external environment) [42] and permeabilisation or disruption of the cell membrane, which allows, respectively, the passage of compounds or the release of intracellular contents (especially potassium, calcium, and sodium ions [34]), adding to the loss of cellular integrity [33,37,43,45,46]. Disruption of the cell membrane may be prompted, for example, by the interaction of phenolics with membrane proteins, inducing alterations in their structure and function, namely in terms of electron transport, nutrient uptake, synthesis of proteins and nucleic acids, and enzyme activity [37]. Additionally, plant extracts may also inhibit DNA and protein synthesis [42], inactivate cellular enzymes (including ATPase) [45,46], and dissipate cellular energy in the form of ATP [33].

Different mechanisms of action have been reported for distinct groups of compounds. In fact, while membrane disruption is associated with the action of terpenoids and phenolics, the antimicrobial properties of phenols and flavonoids seem related to their chelating properties, complexing metal ions that are essential for bacterial growth, whereas coumarin and alkaloids seem to produce effects on genetic material [33,41]. In turn, the antimicrobial activity of some organosulfur compounds, such as onion and garlic isothiocyanates, is due to the inactivation of extracellular enzymes through oxidative dissociation of -S-S- bonds [37]. The mechanism of action may also be dependent on the concentration of the compounds, as it has been shown that at a low concentration, phenols inhibit microbial enzyme activity, whereas at high concentrations, they induce protein denaturation [44].

Irrespective of the mode of action, it is recurrent that Gram-positive bacteria are more susceptible to plant extracts and phenolic compounds than Gram-negative, whose greater resistance is due to the existence of lipopolysaccharides in their outer membranes [37,43,47].

Considering that cheese is a fermented product that contains natural and, sometimes, artificially added microbial populations of LAB, which are a group of Gram-positive organisms, it is reasonable to question if using plants and plant extracts as preservatives may influence bacterial metabolism and/or inhibit this beneficial set of bacteria, potentially compromising the fermentation process. Some studies have reported on this drawback [48,49,50,51], including that of Shori et al., who observed a reduction in peptides content and free amino acids of cheeses in the presence of three different types of plant extracts (*I. verum*, *C. longa*, and *P. guajava*), caused by the impairment of LAB growth and, consequently, LAB proteolytic activity [52].

Nonetheless, the ability of herbal extracts to impact LAB is determined by a number of variables, including the genus, species and strain of the LAB, as well as the plant species and the extraction method used, for example [53]. Various studies have shown that when selected plant extracts are employed in appropriate amounts, they may be able to promote the growth of desired microorganisms, or at least not affect them negatively while avoiding the development of harmful bacteria [53]. For example, Mohamed et al. [54] reported the inhibitory effect of ethanolic and aqueous extracts of *Moringa oleifera* leaves against numerous pathogens in vitro, stressing that these did not inhibit LAB growth. In addition, Ziarno et al. [53] investigated the effect of seven plant extracts (valerian, sage, chamomile, cistus, linden blossom, ribwort plantain and marshmallow) with known antimicrobial activity against pathogens on the activity and growth of LAB and observed that the addition of such extracts up to 3% in milk did not hinder the growth of LAB in fermented milk drinks such as yoghurts. Likewise, Chouchouli et al. [55] supplemented yoghurts but with grape seed extracts and did not observe any effect on pH or the viability of *Lactobacilli*.

Considering the distinct results described in the literature, it is important to establish if a particular plant extract can be successfully used in cheeses by evaluating its impact on the growth and the technological properties of LAB populations, whether they are endogenous raw milk flora or intentionally added starter cultures.

Other issues that should be considered when adding plant extracts to cheeses are, for instance, the loss of bioactive compounds during cheesemaking and storage and the interactions between plant extracts and the cheese matrix, which can have an impact on the texture and organoleptic characteristics of the novel cheese [17]. Even though the sensory attributes of the novel cheese may be different from the corresponding “traditional” cheese right after manufacture, it seems that they do not change drastically during ripening and storage, behaving similarly to cheeses without plant extracts [56,57].

The food matrix is an important factor as interactions with food ingredients occur, resulting in reduced biological activities of the natural compounds when comparing the results of in vitro and in situ (cheese) studies. More specifically, it is generally accepted that high concentrations of lipids or proteins limit the antimicrobial efficacy of plant extracts [58,59,60,61]. Studies regarding the effects of carbohydrates on the antimicrobial activity of plant extracts are scarce [62], as most of the literature focuses on the interaction between carbohydrates and plant essential oils. In this case, different authors report contrasting results: Gutierrez et al. [63] observed a reduction in oregano and thyme essential oils efficacy when testing 5% and 10% starch concentrations, whereas Shelef et al. [64] reported that carbohydrates in foods do not protect bacteria from the antimicrobial action of essential oils, at least not as much as fat and protein. The complexity of the food structure also plays an important role in the biological activity of plant extracts in food, as well as the changing variables during cheese production (namely water activity, pH, microflora composition, temperature and nutrient composition) [43].

Natural compounds can be lost during cheese making or storage as a result of their sensibility to environmental factors (including light, temperature, oxygen and pH [43,65], which can cause the epimerisation of bioactive components [65]), solubility in whey [66] or solubility of hydrophobic active molecules in lipidic phases [18]. Aqueous phases are generally the preferred ones for cell growth [67], not lipidic phases, although some bacteria have been reported to prefer the fat–water interface in emulsion systems [68,69,70].

Although not as intense as essential oils, plant extracts may still negatively affect the sensory characteristics of the food product, especially if the concentrations needed to inactivate pathogens and ensure food safety are higher than those that lead to acceptable sensory properties of the treated products [63]. Tayel et al. [71], for example, evaluated the impact of flavouring with plant extracts on the sensory attributes of processed cheese, and the trained panellists scored the taste and odour of cheeses with extracts of cloves, cress and sage lower than those of the control cheeses. On the other hand, the same study reported enhancements in terms of odour, taste, colour, and overall quality when cinnamon, lemon grass, and oregano extracts were added to the cheese [71].

Other studies have also reported improved sensory quality of cheeses containing plant extracts [72,73,74,75], thus showing that the sensory issue does not always arise and that it is dependent on the type of extract used and the dose applied, as well as other factors. For example, Lee et al. [72] observed that the addition of *Inula britannica* flower extract (0.25% to 1%) did not significantly affect the odour and taste of Cheddar-type cheeses; and Abd El-Aziz et al. [73] also evaluated the flavour of ginger extract-fortified (0.15 and 0.3%) soft cheeses and found no differences compared to the controls. In turn, Mahajan et al. [74] reported significantly higher scores for flavour and texture when applying pine needle extract (2.5% and 5.0%) to cheeses.

To avoid interactions with food components, degradation and loss of bioactive compounds, as well as the unpleasant taste of polyphenols, bio-based functional packaging materials incorporating natural active compounds and ingredients may be used (for example, coatings and edible films using nano- and microencapsulation techniques) [41,43,65].

Other concerns that must be considered include: (i) the effects of plant extracts and their natural compounds on human health, as typical toxicological information such as “acceptable daily intake” or “no observed adverse effect level” are usually not available [33]; and (ii) the economic costs, legislation, and practical effectiveness [43] of using plant extracts as preservatives in the food industry.

The potential toxicity of plant extracts is generally difficult to define when considering the problems in terms of their standardisation due to the great variability in their composition between batches [33]. In terms of economic costs and legislation, it is crucial that the price of natural preservatives is competitive in comparison to that of synthetic compounds providing comparable antimicrobial effect and that plant additions in and on foods comply with the existing legislation [76,77], which nonetheless is still limited and must be improved (for example, natural additives are legislated in the same manner as synthetic ones, making it sometimes difficult to understand how production is carried and what is their source [78]).

Overall, it is clear that plant extracts can be useful as antimicrobial agents in foods, including raw milk cheeses, although further scientific and legal grounds are needed to motivate and simplify the use of such additives.

### 3.2. Lactic Acid Bacteria (LAB)

Traditional raw milk cheeses exhibit a complex microbiota, including LAB naturally occurring in milk and purposefully introduced LAB [79]. They comprise a large and heterogeneous group, and bacterial communities differ vastly among raw milk cheeses, but usually, the main genera identified in raw milk artisanal cheeses include *Lactococcus*, *Lactobacillus*, *Enterococcus*, *Streptococcus* and *Leuconostoc* [79,80].

LAB can be relevant for their role as starter cultures, which promote the rapid acidification of milk (crucial for adequate fermentation and production of high-quality cheeses) through the production of organic acids (primarily lactic and acetic acids) [79]. Starter cultures and adjunct cultures (also called non-starter LAB) can also contribute to the maturation of cheese and the development of desirable texture, flavour, aroma and nutritional value as a result of their metabolic features [79,81]. Various selected LAB strains or mixtures of strains are commercially available as starter cultures for cheese production, and the most frequently used species are *Lc. lactis* (particularly subspecies *lactis* and *cremoris*), *S. salivarius* subsp. *thermophilus*, *L. helveticus*, and *L. delbrueckii* [79,82].

Furthermore, LAB may also have probiotic potential, meaning that they can offer health-promoting benefits to consumers. These include immune system modulation [83], improvement of mental health via the gut–brain axis [83], degradation of nutrient-damaging compounds, such as biogenic amines [84] and cholesterol [85], and increase the quantity of beneficial compounds, such as antihypertensive peptides [86], short-chain fatty acids [87], γ-aminobutyric acid and conjugated linoleic acid [88].

Besides their role in successful fermentations, contribution to textural and sensorial characteristics, and health-promoting properties, some LAB species and strains can also act as antimicrobial agents during and after fermentation throughout the maturation/storage step. This can be due to competition for the adherence site [89], competition for nutrients (i.e., Jameson effect [90]), ability to acidify the environment, and ability to produce antimicrobial metabolites during fermentation, which remain in the final product (except for volatile compounds) [79,91]. In fact, some studies have screened the antimicrobial properties of these microorganisms as a strategy to improve the safety of cheeses and successfully used cocktails of LAB strains to hinder the growth of pathogenic bacteria [91,92,93].

The antimicrobial metabolites produced by LAB that reduce the risk of pathogen growth and survival include organic acids, hydrogen peroxide, diacetyl, fatty acids, reuterin and bacteriocins [79].

Acidification of the environment by organic acids creates adverse conditions for the growth of spoilage and pathogenic microorganisms [94]. *S. aureus*, for example, is strongly inhibited by lactic and acetic acids, as most Gram-negative and neutrophilic bacteria [95]. Undissociated organic acids can diffuse across the cell membrane of pathogens when pH_environment_ < pKa and dissociate within the cell (due to the higher cellular pH), which lowers the cytoplasmic pH [94]. This affects various metabolic processes, promotes the accumulation of toxic anions, dysregulates cell homeostasis, and neutralises the electrochemical proton gradient, disrupting the substrate transport systems and the cell membrane, which potentially leads to the death of the organism [37,79,94]. The concentrations and types of organic acids produced during fermentation are specie- and strain-dependent and also vary with matrix composition and growing conditions [96].

Hydrogen peroxide can be produced by LAB in the presence of oxygen through the action of flavoprotein oxidases or NADH peroxidases [96]. Since LAB cannot degrade this compound, it accumulates in the medium, exerting its bactericidal effect through the destruction of basic molecular structures of cell proteins, denaturation of metabolic enzymes (by oxidation of sulfhydryl groups), and peroxidation of membrane lipids, which increases cell membrane permeability [94,95]. Hydrogen peroxide may also serve as a precursor to the DNA-damaging superoxide ($O_{2}^{•-}$) and hydroxyl (^•^OH) free radicals [94]. In milk, hydrogen peroxide activates the lactoperoxidase system, which has proven bacteriostatic and/or bactericidal activity against various Gram-positive and Gram-negative bacteria [96,97].

Diacetyl is an aromatic compound produced by some LAB strains in the presence of organic acids such as citrate, which is converted via pyruvate into diacetyl (citrate fermentation) [94,96]. Figure 2 displays a schematic representation of the diacetyl production via citrate fermentation.

*Lactobacilli* and *Enterococci* are the genera associated with high diacetyl production, whereas *Leuconostoc* strains produce none or low amounts of diacetyl from citrate [99]. Jay [100] showed that diacetyl was much more effective against Gram-negative bacteria, yeasts, and moulds than against Gram-positive bacteria, while LAB and clostridia were virtually unaffected. The same study also showed that the inhibitory activity of diacetyl against Gram-negative bacteria was related to its interference with arginine utilisation in the periplasmic space, and that pH has an inverse synergistic effect on diacetyl’s bioactivity (lower pH, higher bioactivity) [100], statements corroborated by the research of Tan et al. [95].

LAB can produce various fatty acids that improve the sensory attributes of fermented products while potentially exerting antibacterial and antifungal activity [79]. The antibacterial mechanisms of action of these compounds include DNA/RNA replication inhibition, cell wall biosynthesis inhibition in Gram-positive bacteria, inhibition of protein synthesis, cytoplasmic membrane disruption and inhibition of metabolic pathways [101]. The literature available reports that both unsaturated and saturated fatty acids have antibacterial properties towards Gram-positive and Gram-negative bacteria [101], but that fatty acids with medium and long carbon chains, such as lauric (12C) and capric (10C) acids, provide higher inhibitory effects than short chain fatty acids (<8C) [101,102]. For a scheme of the possible cell targets and mechanisms of antimicrobial action of free fatty acids, please refer to Desbois and Smith [102].

*Lactobacillus reuterin* strains can anaerobically convert glycerol into 3-hydroxy-propionaldehyde (3-HPA), which in aqueous solutions exists in equilibrium as a dynamic system of 3-HPA, 3-HPA hydrate, 3-HPA dimer and acrolein [96,103,104], as depicted in Figure 3. This multi-compound system is commonly known as reuterin [96,103,104].

Effective against Gram-positive and Gram-negative bacteria, yeasts, moulds and protozoa [44], this broad-spectrum antimicrobial aldehyde can also be produced by other LAB, including *L. brevis*, *L. buchneri*, *L. collinoids*, and *L. coryniformis* [105]. The antimicrobial activity of reuterin has been linked to the ability of 3-HPA to cause depletion of free thiol groups in glutathione, proteins and enzymes, resulting in an imbalance of the cellular redox status and leading to bacterial cell death [106]. However, the work of Engels et al. [103] suggested, for the first time, that acrolein, and not 3-HPA, is the active compound responsible for the antimicrobial activity attributed to reuterin, and the proposed mechanism of action is schematically represented in their work [103]. The high potential of reuterin as a food biopreservative is supported by its hydrosolubility, stability over a wide range of pH and temperatures, and resistance to degradation by proteolytic and lipolytic enzymes [37,104]. Moreover, reuterin has a wider range of antimicrobial activity than bacteriocins and other non-bacteriocin antimicrobial compounds [104]. However, due to legislative and regulatory requirements, reuterin is not yet commercially available [107].

To that, bacteriocins are extracellularly released bioactive peptides or peptide complexes synthetised in ribosomes [49]. They have a narrow-to-broad antimicrobial effect against bacteria in the same species or across genera, respectively [108], and the producer cell exhibits specific immunity to the action of its own bacteriocin [94]. The majority of bacteriocins produced by LAB are active only against LAB and other Gram-positive bacteria [109,110], but some studies reported on their effectiveness also against Gram-negative bacteria [110,111]. Antifungal bacteriocins have also been reported, with *Lactobacillus* species being the most predominant isolates associated with such compounds [112]. Bacteriocin-producing LAB include *Lactococcus*, *Lactobacillus*, *Pediococcus*, *Streptococcus* and *Enterococcus* strains [108]. The mechanism of action of bacteriocins depends on their primary structure [111]. In bacteria, while some bacteriocins can promote the formation of pores in the phospholipidic bilayer of the cytoplasmic membrane, causing the dissipation of the proton motive force and loss of cell contents, others can inhibit cell wall synthesis or enter the cytoplasm, and affect gene expression and protein synthesis [111]. The antifungal mode of action of protein compounds by LAB, however, remains somewhat unclear, requiring further studies [112,113]. Bacteriocins maintain activity at high temperatures and over a large pH range, and as they are rapidly hydrolysed in the human gastrointestinal tract by digestive proteases, they pose no negative impacts on the gut microbiota [79]. Currently, and although other LAB bacteriocins have shown potential to be used as biopreservatives, only nisin A, produced by *Lc. lactis*, and pediocin PA-1, produced by *P. acidilactici*, have been approved as food preservatives for industrial application and are commercially available [79,81].

Considering the vast diversity of LAB species and antimicrobial metabolites available, there are numerous possibilities for improving food safety and preventing microbial food spoilage. Nonetheless, it is important to consider any potential limiting factors that might reduce the antimicrobial activity of LAB or its compounds. In this sense, the food matrix and its inherent microflora [114], environmental conditions (such as temperature and pH), aerobic conditions, LAB growth phase and load [89], and pathogen content, for example, are among the factors that should not be disregarded when aiming to use such biopreservatives in foods.

## 4. Thermisation

Thermisation is the standard description for a range of sub-pasteurisation heat treatments of milk, generally from 57 to 68 °C with a holding time between 5 s and 30 min, that is able to reduce bacterial contamination by 3 to 4 log [19,21,115,116,117,118].

Unlike pasteurisation, thermisation causes minimum collateral heat damage to milk constituents, has a mild effect on the raw milk flora and the functionality of milk caseins and salts, and has a reduced impact on the sensory profile of the final cheeses [20,115,116,117,119]. For example, since the heat load is lower compared to that used in pasteurisation, enzymes involved in cheese flavour development, such as lipoprotein lipase, are less inactivated [117]. For this reason, this process may be suitable for the production of artisanal cheeses as it reduces microbial counts and simultaneously enables the profile of the heat-treated milk to be closer to that of raw milk, thus allowing the desirable sensorial properties of typical raw milk cheeses to develop [20].

The mechanisms for heat inactivation of mesophilic microorganisms have been extensively studied, and while the ultimate cause leading to cell inactivation by heat remains uncertain, it is clear that heat can affect a wide range of cellular structures and functions, generally known as cellular targets [120,121]. Focusing on non-sporulating bacteria, the cellular targets most affected by heat treatments are the outer and inner membrane, the peptidoglycan cell wall, the nucleoid, the cell’s RNA, the ribosomes, and the proteins [120].

Damage to the outer cell layers of bacteria (cell wall for Gram-positive bacteria; outer membrane for Gram-negative bacteria) has been reported by several researchers: in Gram-negative bacteria, damage to the outer membrane after mild thermal treatment can be verified by loss of outer membrane lipopolysaccharides [122] and morphological and structural changes [123] in membrane integrity and permeability, which leads to the release of periplasmic proteins and sensitivity to hydrophobic antibiotics, for example [124,125]. The cell wall of Gram-positive bacteria is also susceptible to heat, but these organisms are generally more heat-resistant due to the high content and extent of cross-linked peptidoglycan of the cell wall [120].

Damage to the cytoplasmic or inner membrane (of Gram-positive and Gram-negative bacteria, respectively) by heat injury can be detected through the loss of intracytoplasmic material leaked from the heated cells, including RNA, DNA, proteins, enzymes, amino acids, and potassium ions, for example [126,127,128]. Furthermore, the formation of membrane vesicles and loss of membrane material and integrity after heat treatments have also been reported [120,129,130].

Although DNA has high thermostability [131], less intense heat treatments can still modify the nucleoid structure and damage the DNA molecule during and after the treatment [120,130]. Heat-induced DNA damage is manifested by single- or double-strand breaks, as well as increased mutation frequency in surviving populations after heat exposure [132,133]. Moreover, single-strand denaturation induces the action of deoxyribonucleases, which further degrades DNA via the hydrolysis of its phosphodiester backbone [134].

RNA and ribosomes, on the other hand, are more heat-sensitive than DNA [135]. In that sense, mild temperatures have been reported to cause degradation of ribosomes and ribosomal RNA (rRNA), with associated leakage of substances from the metabolic pool (free amino acids and proteins, for example) [136,137,138] that precedes loss of cell viability. Denaturation of 70S ribosomes and 30S and 50S ribosomal subunits can be a consequence of membrane heat damage and subsequent depletion of magnesium ions from within the cell, as they are essential for the maintenance of the coupled ribosome subunits [136].

Proteins, whether structural or functional (enzymes, for example), may undergo denaturation when bacterial cells are thermally stressed [121]. Protein pumps and channels are also heat-sensitive [120], and, as a response to misfolding and denaturation, protein aggregation may also occur [139]. Rosenberg et al. found a correlation between the thermodynamic parameters of protein denaturation and the death rates of several bacteria [140]. Nevertheless, irreversible denaturation of some proteins might not be lethal to the cell if they can be resynthesised after the heat treatment. On the other hand, it is hypothesised that irreversible denaturation of all copies of RNA polymerase, for example, would represent a lethal event, as this enzyme could not be resynthesised by a cell lacking a single copy [141]. Research has shown that proteins irreversibly denatured by heat are governed by chemical modifications, including deamination of Asn/Gln residues, hydrolysis of peptide bonds at Asp-X residues (X being a small hydrophobic residue), and disulphide bond scrambling [141].

To summarise, the most relevant cellular events that can occur after heat exposure include permeabilisation of membranes, DNA and RNA alterations, loss of ribosome or protein conformation and loss of intracellular components [120]. These events are represented in Figure 4.

As microbial inactivation by heat is a multi-target phenomenon, these events may be interconnected and are likely to occur simultaneously [120]. In any case, the lethality of heat treatment is contingent on the alteration of at least one critical component (one whose destruction triggers cell death) beyond a critical threshold, which can be a result of the direct effect of heat on the critical cellular target itself or a consequence of a parallel alteration of another cellular target [120]. It is also crucial to consider that the resistance of each cell target depends on the environmental conditions and the type of microorganism (pH and water activity of the medium during the heat treatment, for example, Gram-positive vs. Gram-negative bacteria, as mentioned before in this section) [120]. Additionally, exposure to sublethal thermal stresses can mediate adaptive responses in bacteria, including the induction of heat shock proteins, which are determinant for protein folding, repair and degradation, and the prevention of aggregation, thus promoting increased heat resistance and, consequently, bacterial survival [142,143].

Thermisation has been noted for both psychrotroph and pathogen control [30,144,145]. Nevertheless, and as previously described, different microorganisms may respond differently to heat treatments, depending on a variety of factors [19]. In this sense, a few authors have reported the survival of some yeasts [146], that some pathogens may remain viable at the lower end of the thermisation temperature range, where the lethal effect is more reduced [30,117], and that thermisation may not be enough to significantly reduce the population of vegetative cells of the more heat-resistant bacterial species (*Enterococcus*, for example) [117,145]. Besides the possibility of some bacteria remaining viable in thermised milk, other shortcomings associated with this thermal treatment are the possible germination of spores present in milk during subsequent cold storage (for example, thermisation at 65 °C for 10 s may be sufficient to stimulate the germination of *B. cereus* spores [115]) and the possible selection for heat-resistant microorganisms such as *M. tuberculosis* and *C. burnetii*, by enabling their survival while reducing competitive flora [2,116]. Thermisation may also have a negative impact on LAB strains and the biodiversity of raw milk bacteria. To this, Sameli et al. [119] observed that thermisation at 60 °C for 30 s reduced the total number of *Leuconostoc*, *Lactococcus* and mesophilic *Lactobacillus*, while producing an enterococcal selecting effect. To avoid such negative effects, it is important that thermisation parameters are carefully selected, aiming to target pathogens while preserving LAB as much as possible. Moreover, the addition of a starter culture post heat treatment to counteract the reduction in LAB numbers may also be recommended.

## 5. Conclusions

Artisanal raw milk cheeses may pose health risks to consumers, considering that the manufacturing processes are not standardised, and good manufacturing practices are not always followed, which can lead to undesirable microbiological quality of the cheeses. To avoid pasteurisation and the use of chemical preservatives, which are unfit for this niche product, this work collected and discussed the main antimicrobial action mechanisms, bacterial targets, advantages, limitations and, whenever possible, relevant commercial applications of two biopreservatives, plant extracts and lactic acid bacteria, as well as a mild heat treatment of milk, thermisation, with the goal of promoting their use in cheese production. The literature currently available is supportive of the use of these strategies for the improvement of the microbiological quality of artisanal raw milk cheeses, although some considerations, such as their impact on the sensory characteristics of the product and on the natural microflora, must be carefully assessed, as referred in this review.

## Figures and Tables

**Figure 1 foods-12-03206-f001:**
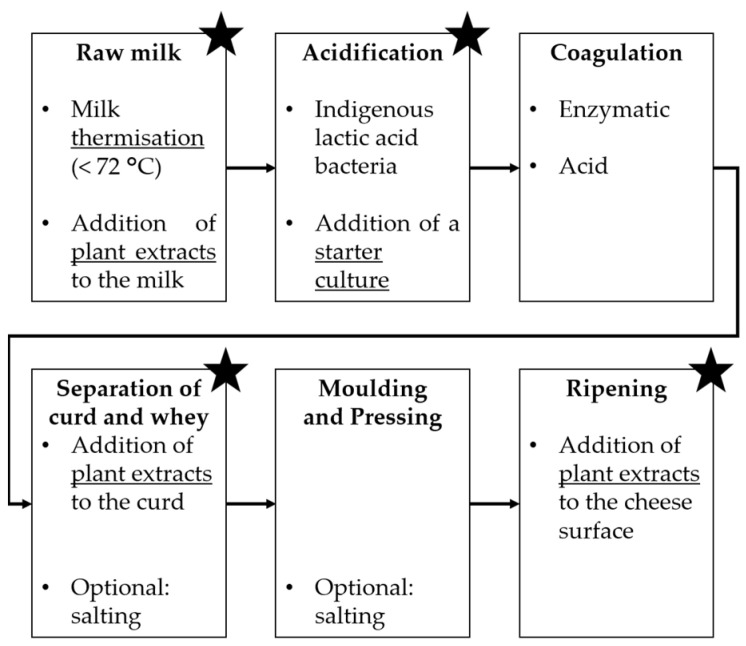
General cheesemaking process. Star symbols indicate at which steps thermisation, addition of plant extracts and/or addition of a starter culture may be implemented.

**Figure 2 foods-12-03206-f002:**
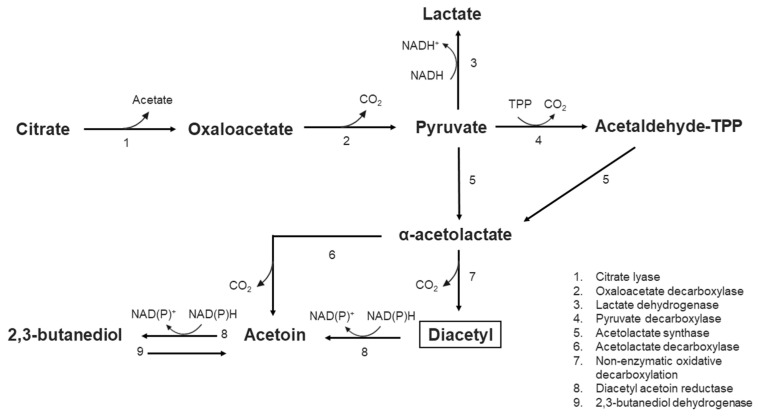
Diacetyl synthesis via citrate metabolism in lactic acid bacteria. TPP: thiamine pyrophosphate. Based on García-Quintans et al. [98].

**Figure 3 foods-12-03206-f003:**
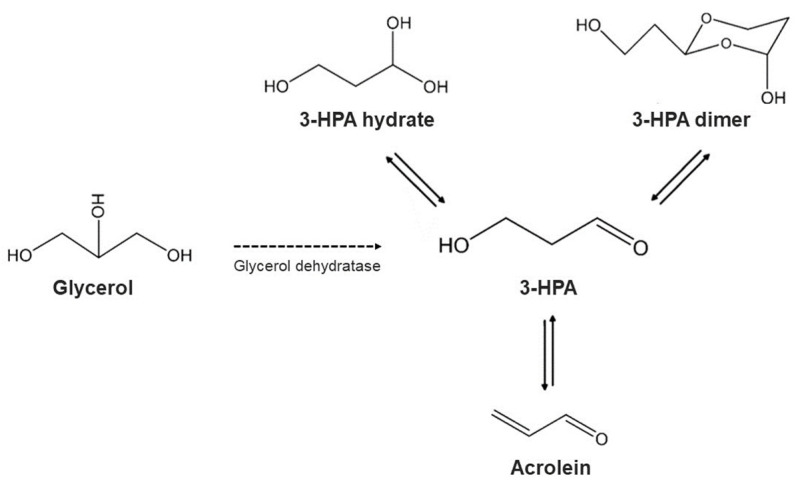
Formation of the reuterin system (comprised of 3-HPA, 3-HPA hydrate, 3-HPA dimer and acrolein) as a result of the dehydration of glycerol by the enzyme glycerol dehydratase. 

 Enzymatic reaction; ⇄ Equilibrium reactions.

**Figure 4 foods-12-03206-f004:**
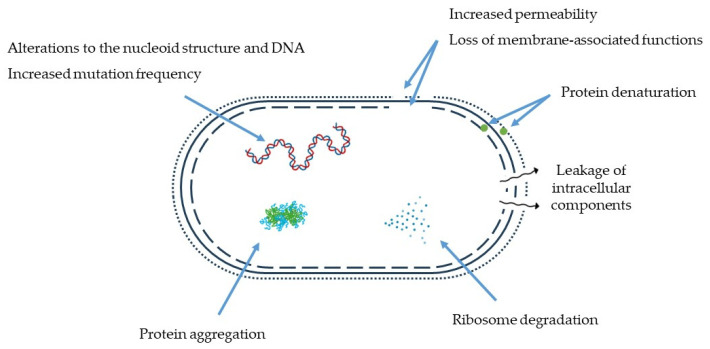
Main cellular events that occur in a vegetative bacterial cell (Gram-positive or Gram-negative) after heat exposure. Based on Cebrián et al. [120].

## Data Availability

No new data were created or analysed in this study. Data sharing is not applicable to this article.

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
