# Peer review of "Mild Heat Treatment and Biopreservatives for Artisanal Raw Milk Cheeses: Reducing Microbial Spoilage and Extending Shelf-Life through Thermisation, Plant Extracts and Lactic Acid Bacteria"

_foods, 2023, doi:10.3390/foods12173206_

Round 1

Reviewer 1 Report

The manuscript entitled “Mild heat treatment and biopreservatives for artisanal raw milk cheeses: reducing microbial spoilage and extending shelf-life through thermisation, plant extracts and lactic acid bacteria” summarizes some of useful methods for improving the quality and shelf-time of raw milk artisanal cheeses. The summary is useful, but the following comments should be carefully considered.

Line 29-32: references should be offered here.

Line 33-40: the unique flavor of raw milk cheeses should be offered here when compared with other kinds of cheeses.

All the names of the species in the manuscript should be italic.

Part 2 “2. Spoilage microorganism in raw milk and raw milk cheeses”, a detailed discussion of the negative effects of the species will offer more useful information for readers. Maybe a table could be offered here as there is even no one figure/Table in the manuscript.

Line 394, rewrite the sentence.

Part 3, a figure of the metabolic pathway of the anti-bacterial chemicals such as fatty acids, diacetyl, 3-HPA, 3-HPA hydrate, 3-HPA dimer and acrolein could be provided.

In addition to exploring appropriate preservation methods for products, analysis should also be conducted on the production process. Due to the fact that microorganisms are naturally inoculated throughout the production process, it is possible to learn from the manufacturing methods of other traditional fermentation foods, such as the combination of exogenous functional strains, to achieve batch stability and quality control of fermentation products and ensure product color and flavor. Such information should be offered in the Introduction or Conclusions section to make the review of the paper more comprehensive, not only a discussion of lactic acid bacteria. Some of recently published papers could provide some references, such as Foods 2023, 12(4), 735; https://doi.org/10.3390/foods12040735, and Foods 2023, 12(3), 644; https://doi.org/10.3390/foods12030644.

Line 378-424, a figure summarizes the mechanism of heating on cell damage could be provided here.

Moderate editing of English language required

Reviewer 2 Report

The authors propose alternative methods to address these issues without resorting to pasteurization or chemical preservatives, which are apparently deemed unsuitable for this product category. Specifically, the article delves into the realm of biopreservatives, namely plant extracts and lactic acid bacteria, along with thermisation. The authors appear enthusiastic about promoting these methods for cheese production with the aim of improving microbiological safety; however real-world cheesemaking practices must be considered.

Line 33: Provide some market tends regarding the artisanal raw milk cheeses’ and the industrial need for biopreservatives, namely plant extracts and lactic acid bacteria, thermisation.

Line 64: Briefly discuss the cost considerations about doing this

Line 97: Include the common processing techniques available help preventing the spoilage of yeast, mold, and other pathogens

Line 192: explain further how it effects the sensory attributes of cheese after production and during storage

Line 239: Explain if plant extracts influence (texture, melt etc.) other properties when this cheese is used as an ingredient

Line 272: Further explain the exact mechanism involved

Line 461: Include a paragraph on how other technologies like HPP and ultrasound and how it could be also potentially employed for this

Line 474: One glaring gap in the discussion is the potential impact of these proposed strategies on the sensory attributes of the cheeses. The article merely hints at "considerations" related to sensory characteristics, leaving readers curious about potential alterations in taste, texture, and overall eating experience. Such omissions raise questions about the holistic feasibility and desirability of the suggested interventions.

In conclusion, this article raises valid concerns about the health hazards posed by artisanal raw milk cheeses due to suboptimal manufacturing practices. The proposed alternative strategies offer a glimmer of hope, but their real-world applicability and their implications for the sensory quality of the end product remain unexplored territories. As it stands, while the article starts a crucial conversation, it falls short of providing a comprehensive and convincing roadmap for the future of artisanal raw milk cheese production.

No specific comments. 

Reviewer 3 Report

Dairy products are highly perishable and contains microbes which are both beneficial as well as toxic cultures. Cheese is a highly cherished product which has its own characteristic aroma flavor, mouth feel and so on .Milk is the basic component for making cheese. Milk is subjected to various processing conditions to make it into a desirable product like cheese. Therefore quality and safety of milk becomes very crucial in cheese making. Artisanal raw milk needs to be protected to prevent its spoilage by undesirable microbes. To achieve this objective various procedures can be adapted as discussed in this review. These include mild heat treatment known as Thermisation .One can also use plant derived materials having anti-microbial properties. In addition , the inherent Lactic acid bacteria can provide protection. The ultimate goal of these approaches is to provide safe product with desirable quality and microflora. The draw back of using pasteurization for this purpose is also high lighted in this review.

Suggestions :

1. The review has become more narrative. A schematic flow diagram of cheese making and indicate at what step the bio-actives from plant extracts need to be added.

2. Prepare a table and indicate where intervention was done to improve the shelf life of the dairy product. How much of improvement in the shelf life of raw materials and sensory properties of final product was achieved should be mentioned.

3. What was the minimum concentration of additives added to improve quality of cheese prepared to be discussed by taking the specific examples from literature.

Minor editing are required to improve the clarity of presentation

Round 2

Reviewer 1 Report

The manuscript can be accepted in the current form.

Reviewer 2 Report

No further comments. 

No further comments.